# Inter-identity amnesia for neutral episodic self-referential and autobiographical memory in Dissociative Identity Disorder: An assessment of recall and recognition

**Rosemary J. Marsh**[1,2], **Martin J. Dorahy**[1,3]*, **Chandele Butler**[1], **Warwick Middleton**[1,3], **Peter J. de Jong**[2], **Simon Kemp**[1], **Rafaele Huntjens**[2]

**1** School of Psychology, Speech and Hearing, University of Canterbury, Canterbury, New Zealand, **2** Department of Clinical Psychology, University of Groningen, Groningen, Netherlands, **3** The Cannan Institute, Belmont Private Hospital, Brisbane, Australia

* martin.dorahy@canterbury.ac.nz

**Data Availability Statement:** Data are available at: https://osf.io/u4r6k/.

## Abstract

Amnesia is a core diagnostic criterion for Dissociative Identity Disorder (DID), however previous research has indicated memory transfer. As DID has been conceptualised as being a disorder of distinct identities, in this experiment, behavioral tasks were used to assess the nature of amnesia for episodic 1) self-referential and 2) autobiographical memories across identities. Nineteen DID participants, 16 DID simulators, 21 partial information, and 20 full information comparison participants from the general population were recruited. In the first study, participants were presented with two vignettes (DID and simulator participants received one in each of two identities) and asked to imagine themselves in the situations outlined. The second study used a similar methodology but with tasks assessing autobiographical experience. Subjectively, all DID participants reported amnesia for events that occurred in the other identity. On free recall and recognition tasks they presented a memory profile of amnesia similar to simulators instructed to feign amnesia and partial information comparisons. Yet, on tests of recognition, DID participants recognized significantly more of the event that occurred in another identity than simulator and partial information comparisons. As such, results indicate that the DID performance profile was not accounted for by true or feigned amnesia, lending support to the idea that reported amnesia may be more of a perceived than actual memory impairment.

## Introduction

Dissociative Identity Disorder (DID) involves a person reporting the existence of at least two separate identities with distinct perceptions, cognitions, and interactions with the environment [1,2]. Recurrent gaps in the recall of experienced events and personal information is a core diagnostic criterion for DID, and can present as two-way amnesia (with no transfer of information reported across identities) or one-way amnesia [with only one identity reporting

**Funding:** The authors received no specific funding for this work.

**Competing interests:** The authors have declared no competing interests exist.

access to the memories of the other; 3,4]. It is not uncommon for adults with DID to present with identities who report full awareness of traumatic experiences from the past, and others that report little or no recall of previous traumatic events [5,6].

Not yet understood is the nature of inter-identity amnesia in DID. Preliminary research suggested that only memories retrieved explicitly exhibited inter-identity impairment [7,8]; however, studies using more objective measures have found both explicitly and implicitly retrieved memories in DID exhibit transfer. It is also evident that although amnesia pertaining to both neutral stimuli and stimuli related to sexual and physical abuse is subjectively reported, both types of memory content exhibit transfer when assessed via objective measures [9–17]. Recent research has also indicated retrieval of autobiographical stimuli [18], and episodic self-referential material [19], similar to the memory transfer pattern displayed by non-self-referential material. The current studies investigated retrieval by means of recall and recognition tests across amnesic identities for episodic self-referential and autobiographical information. The episodic memory system holds information about experienced events and when retrieved these memories are paired with autonoetic consciousness, which reflects the process of recollecting and re-experiencing the event [20]. If a memory does not elicit autonoetic consciousness, it is retrieved from the semantic memory system, and paired with knowledge that the event occurred without the rich recollective experience (noetic consciousness; 20). Although Huntjens et al. [14] found no differences for information being"remembered" (autonoetic) or "known" (noetic) for amnesic identities, the research did not utilise episodic self-referential and autobiographical stimuli which may have resulted in limited access to autonoetic consciousness.

## Study 1

The aim of the first study was to assess explicit retrieval of self-referential memories across DID identities that report amnesia for each other. To assess the extent of amnesia, several comparison samples were included. We recruited two non-clinical comparison groups, one that showed no amnesia and a second that was truly unaware of the self-referential material assessed (i.e., the material was not shown to them). We also included a group of people instructed to consciously simulate DID to address the suggestion that DID may be a simulated disorder [21]. The primary researcher was blind to whether DID and simulator participants truly had DID or were feigning it as a simulator. Thus participants were from four samples: 1) DID participants reporting two-way amnesia, 2) simulator participants educated on how to mimic DID, 3) non-clinical comparison participants given stimuli shown to one identity in the DID and simulator groups (i.e., half of the stimuli; partial information group), and 4) non-clinical comparison participants given stimuli shown to both identities in the DID and simulator groups (i.e., all of the stimuli; full information group).

Based on the DSM-5 criteria of recurrent gaps in memory, the following hypotheses were tested: (1) For free recall, DID participants would exhibit amnesia for vignettes encoded in a different identity, a similar profile to the simulator and partial information non-clinical comparison groups. Full information comparison participants would exhibit comparable memory retrieval for both vignettes; (2) For forced choice recognition, a similar pattern of results was predicted; (3) In case of (partial) transfer between identities, DID participants would exhibit relatively higher remember (autonoetic) scores for vignettes encoded in the same identity and relatively higher know (noetic) scores for vignettes encoded in the other identity. Full information participants would exhibit comparable remember-know scores across vignettes.

## Material and methods

### Participants

**DID sample.** Nineteen DID participants were recruited from referrals via clinicians at a dedicated hospital-based programme in Australia. Participants were deemed stable enough to commence the study by their treating clinician. Inclusion criteria were, (1) a pre-existing DID diagnosis; (2) a confirmation of the DID diagnosis via the Dissociative Disorders Interview Schedule (DDIS) administered by the primary researcher; (3) the capacity to engage two identities who report a lack of memory for events experienced by the other identity (two-way amnesia); and (4) the ability to switch between these two identities on request. Participants were excluded if they, (1) were too impaired by their psychiatric state to concentrate on the computer tasks; (2) were unable to switch between identities or retain an identity in executive control; or (3) did not report two-way amnesia between participating identities. Participants were told that the study would examine memory in different identities in a DID sample. They self-selected the two identities (labelled A and B) that took part in the study with the instruction to choose one identity with more awareness of past distressing experiences and one identity with less awareness of autobiographical distressing experiences. They chose which identity took part first (as identity A) and second (as identity B), and awareness of distress was not explored in this study. DID participants received a $20 shopping voucher for participating. Of the 19 DID participants tested, seven were removed for failing to complete the task ($n = 1$), being unable to switch between identities ($n = 3$), or experiencing test interference from a third identity ($n = 3$). The final DID sample contained 12 participants.

**Comparison samples.** Forty-one comparison participants took part in the study. These were members of the general population of Christchurch and Brisbane recruited by snowball sampling ($n = 26$) to match the mean age and gender of DID participants, and undergraduate psychology students ($n = 15$) recruited via a participant research pool or advertisements. Comparison participants reported no memory or attentional deficits. They were randomly assigned to a 'partial information' group or a 'full information' group. Participants in the partial information group (half-stimuli group) received the memory stimuli presented to identity A in the DID group, while participants in the full information group received the memory stimuli given to both DID identity A and identity B. Comparison participants were not aware that the study was researching DID. Participants received $20 in shopping vouchers for participating.

**Simulator sample.** Sixteen DID simulator participants took part in the study. They were professional and amateur actors from a University Theatre and Film Department, and various local theatre companies. They developed two identities that they were instructed to have amnesia between them, and followed the same protocol as the DID sample. Of the sixteen simulator participants tested, two were removed (both female) for declining to complete the full test session due to the length of the study being greater than they anticipated, leaving a sample size of 14. Participants received $20 in shopping vouchers for taking part. Ten simulator participants were not administered the DDIS due to this questionnaire being added to the procedure following their participation (thus $n = 4$ for the DDIS).

Table 1 displays the demographic details for each sample. The groups differed significantly for age, $F(3, 63) = 3.46$, $p = .02$, $\eta_p^2 = .14$, with simulators significantly younger than partial information comparisons ($p = .03$). DID participants showed a trend to differ from simulators ($p = .078$) and did not differ significantly from partial information ($p = 1.00$) or full information comparisons ($p = 1.00$). Simulators showed a trend to differ from full information comparisons ($p = .06$), while partial information and full information comparisons did not differ significantly ($p = 1.00$). There were minimal variations in gender, but due to the low count for males across groups, inferential statistics were not conducted. Comparative analysis was also not utilized for ethnicity and education due to cells with low counts.

**Table 1. Participant demographic data across groups.**

| | DID (*n* = 12) | Simulator (*n* = 14) | Partial Information Comparison (*n* = 21) | Full Information Comparison (*n* = 20) |
|---|---|---|---|---|
| Age *M (SD)* | 39.17 (3.39) | 28.86 (3.85) | 39.10 (1.90) | 38.35 (1.62) |
| Gender *n* (%) | | | | |
| Male | 0 (0%) | 2 (14.3%) | 1 (4.8%) | 1 (5%) |
| Female | 12 (100%) | 12 (85.7%) | 20 (95.2%) | 19 (95%) |
| Ethnicity* | | | | |
| New Zealand European | 0 (0%) | 12 (85.7%) | 11 (52.4%) | 16 (80%) |
| Māori | 0 (0%) | 0 (0%) | 0 (0%) | 0 (0%) |
| Australian European | 10 (83.3%) | 2 (14.3%) | 1 (4.76%) | 0 (0%) |
| European Other | 3 (25%) | 1 (7.14%) | 5 (23.8%) | 0 (0%) |
| Other | 1 (8.3%) | 0 (0%) | 5 (23.8%) | 4 (20%) |
| Qualification§ | | | | |
| High school certificate | 4 (33.3%) | 7 (50%) | 2 (9.5%) | 2 (10%) |
| Post-high school non-university (e.g. Trade certificate) | 1 (8.3%) | 0 (0%) | 8 (38.1%) | 7 (35%) |
| University Level | 5 (41.7%) | 7 (50%) | 11 (52.4%) | 11 (55%) |

*Note.*

* For the dissociative identity disorder (DID) group, two participants chose two ethnicities. For the simulator and amnesic groups, one participant chose two ethnicities. For the nonamnesic comparison group, one participant chose two ethnicities, and one did not indicate an ethnicity.

§ For the DID group, two participants reported no qualifications.

## Materials

All participants completed a vignette memory task and a questionnaire battery.

**Vignettes.** Two vignettes were developed for the study. These vignettes expanded on previous work that has used stories in a laboratory context with clinical and non-clinical participants [22]. The topics of the two vignettes focused on being at home and in a park and had neutral emotional content. The home story involved participants surfing the internet and telling a friend about clothing specials they had found. The park story involved participants walking around a park, feeding the ducks and it starting to rain. Both vignettes were 14 sentences long. Vignettes can be viewed in full at https://osf.io/u4r6k/.

**Questionnaires.** All participants completed a demographic questionnaire assessing age, gender, ethnicity and education level, as well as the Dissociative Experiences Scale (DES), the PTSD Symptom Scale—Self-Report (PSS-SR) and a vignette emotion rating scale. DID and four simulator participants were also administered the DDIS, which was presented first in the initial test battery. Two patients were not administered the DDIS due to an administration error. All 10 participants who were administered the DDIS had their diagnosis confirmed. The administration order of the DES and PTSD-SR was counterbalanced across participants and the vignette emotion rating scale was presented directly after each vignette was administered. Finally, the researcher assessing participants completed a Post-Experiment Questionnaire assessing whether they felt each participant had DID or was a simulator.

**DDIS [23].** The DDIS is a structured diagnostic interview for dissociative disorders. The DID section was used to make an independent, structured clinical interview diagnosis of DID. Respondents answer questions relating to diagnostic criteria for DID using 'yes', 'no' or 'unsure' responses. The DDIS is a psychometrically sound measure assessing the presence of dissociative disorders and has good inter-rater reliability [24].

**DES [25].** The DES is a 28 items self-report measure assessing dissociative experiences and symptoms on an 11 point scale from 0 (never) to 100 (always) [25]. The DES has excellent psychometric properties in clinical and non-clinical populations [25–27]. Internal consistency was adequate across all samples in the current study (Cronbach alphas > .85).

**PSS-SR [28].** The PSS-SR was included to assess PTSD symptoms, which are often evident in DID [29,30]. The PSS-SR is a 17-item self-report measure assessing on a scale from 0 (not at all) to 3 (five or more times per week/very much/almost always) the presence and severity of PTSD symptoms [28]. The PSS-SR has good psychometric properties [28] and the Cronbach alphas in the current study were all above .87.

**Vignette task emotion ratings.** Directly after each vignette was administered, participants rated how much the information activated feelings of shame, disgust, anxiety, embarrassment, guilt (compiled into an overall negative emotion score), and happiness along a 0 (not at all) to 100 (completely) point Likert scale. The emotion ratings were included to ensure that the vignettes did not differ in emotional tone.

**Post-experiment questionnaire.** A post-interview questionnaire was included for the primary researcher to rate whether DID and simulator participants appeared to present with features of DID. The researcher completed a questionnaire for each participant to determine whether they appeared to have genuine or feigned DID symptoms (e.g., amnesia), which assessed multiple facets of their presentation such as whether they appeared to have distinct identities and in what ways their appearance differed across the tasks. The researcher also rated whether they appeared to have amnesia between the identities and whether the person appeared to have DID. Responses were made on either a likert (not at all–completely) or categorical (yes, no, questionable) scale. This assessment was included to determine if the simulators could successfully mimic DID (i.e., have the researcher believe they had the disorder).

## Experimental measures

**Free recall (FR).** *FRImmediate*. Immediately after hearing each vignette, participants were given two minutes to state all the details they could remember. They were prompted following their retrieval effort with the questions "are those all the details that you can remember?" and then "does anything else come to mind?" The FRImmediate acted as a memory consolidation task.

*FRDelayed*. Once both vignettes had been administered and following an approximate 45 minute delay, participants were asked to state the details of the two vignettes. Participants were prompted after initial retrieval with, "are those all the details you have about that story?". Following their recall of the first vignette, if they did not spontaneously recall the other vignettes (e.g., DID participants, simulators, partial information comparisons), participants were prompted with "did you hear any other stories?" An affirmative response brought an invitation to outline that story. A negative response resulted in the researcher administering an additional prompt of: "do you remember anything about being in your house [or the park]?" If the response was negative, the researcher administered an additional prompt of: "do you remember anything about being on your computer [or feeding ducks]?" These prompts were designed to assess if DID and simulator participants could retrieve and volunteer information given to their amnesic identity. If an affirmative response was received to any prompts participants were asked to recall everything they could about that vignette. The FRDelayed assessed free recall memory of the vignettes and acted as a marker of retrieval interference to assess if one vignette's details were present in the other story and if the DID and simulator participants could retrieve information given to their 'amnesic' identity.

**Forced choice recognition.** *Stimuli Recognition.* Participants were presented with 38 sentences individually placed in the middle of a computer screen. Half of these were related to each neutral vignette (19 in total; i.e., ten and nine sentences), and half were not related to either vignette (19 distractor stimuli). Distractor sentences were matched to the vignette stimuli based on word category (i.e., noun, verb; e.g., target sentence: "did you walk up to the duck pond"; distractor sentence: "did you walk up to the pet store"). Participants indicated whether they recognized the sentences as representing details that had happened in the stories by pressing the Y key on the keyboard, or if they did not recognize them, by pressing the N key. Order of presentation of the stimuli was randomised across each participant. The forced choice recognition task was used to assess recognition memory of the vignettes. It was administered after the FRDelayed task.

*Remember/know.* If participants indicated they recognized the sentence as representing a detail from the vignettes, they were asked to state whether they either 'remembered' or 'knew' that it had happened. A 'remember' response was explained to participants as being paired with an actual recollection of the event, while a 'know' response was explained as being paired with only a feeling that the event happened [24]. Participants pressed the R key on the keyboard to indicate 'remember', and the K key to indicate 'know'. This task assessed autonoetic and noetic memory retrieval.

## Procedure

The study was part of a larger experiment on memory transfer in DID requiring written informed consent from all participants [19] and was approved by both the University of Canterbury's Human Ethics Committee and the Belmont Private Hospital's Medical Advisory Committee.

**Simulator training phase.** First, participants rated their knowledge of DID from very knowledgeable to no knowledge (https://osf.io/u4r6k/) and then watched a video describing DID that provided information to aid simulator performance. The video was 8.27 minutes long and consisted of clips from two documentaries "Multiple Personality Disorder: The Search for Deadly Memories" [31], and the trailer of "When the Devil Knocks" [32]. Both clips presented people with DID, and information about how and why it can develop. A Dissociative Identity Disorder Information Sheet was also provided and consisted of three pages of information outlining the DSM-5 diagnostic criteria of DID and answers to frequently asked questions (e.g., how do the identities develop, is it obvious when a person switches personalities, what are the symptoms of DID). The information sheet was adapted from one provided by the Sidran Foundation. An Identity/Character Description Sheet was also administered to simulators which required participants to complete 18 questions about their created identity (e.g., age, temperament). Participants were given education on how to mimic DID from a professional actor which included practice instructions on how to create and switch between identities. Participants in this group were told the researcher conducting the assessment session was blinded to their simulator status, and they needed to have them believe that they had a DID diagnosis and experienced amnesia between identities. To enhance motivation to effectively simulate DID, simulator participants were informed that the person most convincingly imitating DID based on the primary researcher's ratings on the post-experiment questionnaire would receive a $50 shopping voucher. Participants were given one-to-two weeks to practice cultivating their dissociative identity before testing.

**Experimental procedure.** The study was separated into five phases: (1) questionnaire battery completion, (2) vignette task presentation and initial free recall, (3) delayed free recall, (4) forced choice recognition task, and (5) post-experiment questionnaire completion.

Phase 1. After consent, the primary researcher administered the DDIS. Participants then completed a demographic information questionnaire, the DES and PSS-SR using the Qualtrics survey platform. The order of the latter three questionnaires was randomized.

Phase 2. Participants completed the vignette task where they listened to one (partial information sample) or two (DID, simulators, full information samples) vignettes, depending on their group allocation. The order of vignette presentation was counterbalanced across participants. (Eight of the final DID sample received the house vignette in identity A, and four received the park vignette in identity A. Eight of the final simulator sample received the house vignette in identity A, and six received the park vignette in identity A. Eleven partial information comparisons received the house vignette in identity A and ten received the park vignette in identity A. Ten full information comparisons received the house vignette in identity A and ten received the park vignette in identity A. Although equal numbers of each group were assigned to have the house and park vignettes presented in identity A, the final allocation was not equal due to some participants in the DID and simulator groups not being included in the final sample).

- DID and simulator participants listened to one vignette in identity A and then switched to identity B where they listened to a second vignette.

- Partial information comparisons listened to only one vignette.

- Full information comparisons listened to two vignettes.

Participants were told to remember as many details as possible and become absorbed in the stories. The vignettes were played through headphones and after each sentence, participants were asked to repeat aloud what had been said, changing from second to first- person perspective (e.g., heard, 'you are in your house', repeated, 'I am in my house'). This step was included to increase the self-referential quality of each story [22]. After each participant had finished repeating the sentence, the next sentence was presented. After each vignette, the immediate free recall task was administered. Participants were then required to complete the vignette emotion ratings questionnaire. DID and simulator participants then switched into their second identity and were administered the second vignette, FRImmediate and emotion ratings. Full information comparisons were presented with the second vignette after receiving the first and completing the FRImmediate.

Phase 3. Following an interval of approximately 45 minutes, the participants completed the FRDelayed task. DID and simulator participants were assessed in the same identity that heard the first vignette (Identity A). They were then asked to recall information from any other vignettes they heard.

Phase 4. Participants were required to complete a forced choice recognition task, in which they had to state whether they recognized sentences presented on a computer screen as representing events that had happened in the two vignettes. If they offered an affirmative response, participants were required to state whether they "remembered" or "knew" that these had happened. DID and simulators were tested in Identity A.

Phase 5. The primary researcher completed the post-experiment questionnaire for DID and simulator participants.

## Data analysis

**Recall.** The number of words accurately recalled from each vignette was compared. Identity A vignettes were contrasted with identity B vignettes. The proportion of correct stimuli (correct stimuli divided by the total number of vignette stimuli) was calculated. Participant

responses to two segments from the park vignette were removed from the recall analysis due to them being rated as too similar to other items in that vignette that remained in the analysis.

**Recognition.** Recognition hit rates were calculated for each vignette (correctly recognized vignette 1 stimuli divided by total number of vignette 1 stimuli; correctly recognized vignette 2 stimuli divided by total number of vignette 2 stimuli). Participant responses to one segment from the park vignette were removed from the recognition analysis due to it being rated as too similar to another item in that vignette that remained in the analysis. False alarms, sensitivity and response bias for each vignette were also calculated. Sensitivity ($d'$) and response bias ($c$) were calculated using z-score procedures outlined in MacMillan and Creelman [33]. $d'$ refers to the proportion of stimuli presented to the participants that were coded as previously seen (hits), while correcting for distractor stimuli that were incorrectly coded as being previously seen (false alarms) ($d' = z(H)–z(F)$). Both identity A and B vignette stimuli were coded as targets and both identities shared the same set of distractor stimuli. Positive scores indicate an ability for participants to distinguish between words previously seen and distractors, while scores within the negative range indicate recognition of distractor stimuli at a higher rate than words seen. A score of 0 indicates no ability to discriminate between words seen and distractor stimuli. Response bias ($c$) is conveyed as a participant's likelihood to respond yes or no to stimuli as having previously been seen and is represented using the midpoint between $z(F)$ and $z(H)$ ($c = -0.5[z(H) + z(F)]/2$). A higher response bias indicates that participants were more conservative (i.e., less inclined to recognize items as old/seen).

**Remember/Know.** A remember/know rate was calculated for each vignette dividing information classified as remembered or known by the hits per vignette.

Data for study 1 can be found at: https://osf.io/u4r6k/

## Results

### Questionnaires

All DID participants and those simulators who received the DDIS were positive for DID. Dissociative experiences measured by the DES differed significantly across groups, $F(3, 63) = 57.95$, $p < .001$, $\eta_p^2 = .734$, with DID participants ($M = 61.16$, $SE = 13.02$) scoring higher than simulators ($M = 44.16$, $SE = 21.31$; $p = .01$), partial information comparisons ($M = 11.34$, $SE = 7.77$; $p < .001$) and full information comparisons ($M = 10.89$, $SE = 8.39$; $p < .001$). Simulators also presented a significantly higher score than partial information ($p < .001$) and full information ($p < .001$) comparisons. Comparison participants did not show a significant difference ($p = 1.00$).

PTSD symptoms measured by the PSS-SR differed significantly across groups, $F(3, 63) = 42.94$, $p < .001$, $\eta_p^2 = .67$. DID participants ($M = 42.67$, $SE = 9.19$) reported higher scores than simulators ($M = 21.93$, $SE = 10.06$; $p < .001$), partial information comparisons ($M = 7.19$, $SE = 7.01$; $p < .001$) and full information comparisons ($M = 11.30$, $SE = 10.48$; $p < .001$). Simulators also presented a significantly higher score than partial information ($p < .001$) and full information ($p = .01$) comparisons. Partial and full information comparison participants did not differ ($p = .63$).

### Vignette emotion ratings

For vignette emotion ratings a mixed measures ANOVA on Vignette (home, park) by Group (DID, simulator, full information comparison) was conducted (Table 2). The partial information comparisons received only one vignette so were not included in the analysis. The home and park vignettes did not differ significantly on ratings of happiness, $F(1, 43) = .02$, $p = .88$, $\eta_p^2 < .01$, or negative emotion, $F(1, 43) = 2.50$, $p = .11$, $\eta_p^2 = .06$. There were also no

**Table 2. Vignette-emotion rating means (with SD).**

| | DID ($n$ = 12) | Simulators ($n$ = 14) | Full Information Comparisons ($n$ = 20) |
|---|---|---|---|
| **Home Vignette** | | | |
| Happiness | 30.83 (27.79) | 30.71 (20.93) | 33.50 (30.31) |
| Negative Emotion | 29.50 (25.41)[a, b] | 5.14 (6.41) | 2.80 (8.45) |
| **Park Vignette** | | | |
| Happiness | 22.50 (19.60) | 42.14 (29.14) | 32.50 (31.27) |
| Negative Emotion | 37.83 (28.62)[a, b] | 11.71 (22.65) | 2.50 (5.87) |

Note

a = DID-Simulator difference, $p < .05$

b = DID-full information comparison difference, $p < .05$.

differences across groups for happiness, $F(2, 43) = .58$, $p = .56$, $\eta_p^2 = .03$; however there was a significant difference across groups for negative emotion, $F(2, 43) = 19.82$, $p < .001$, $\eta_p^2 = .48$. DID participants rated both vignettes as significantly more negative than simulators ($p < .001$) and full information comparisons ($p < .001$). Simulators did not differ significantly from the full information comparisons ($p = .55$). There was no significant difference in DID participants' ratings between both vignettes, $t(11) = -.96$, $p = .36$, indicating that both vignettes were rated as having a similar negative emotional connotation.

## Post-experiment simulator performance questionnaire

DID participants and simulators did not differ significantly based on the experimenter ratings of the presence of amnesia between identities, $F(1, 23) = 1.36$, $p = .26$, $\eta_p^2 = .06$, and changes in appearance based on affect, $F(1, 23) = 1.09$, $p = .38$, $\eta_p^2 = .05$, body posture, $F(1, 23) = .64$, $p = .43$, $\eta_p^2 = .03$, and voice characteristics, $F(1, 23) = .97$, $p = .34$, $\eta_p^2 = .04$. There was a trend for groups to differ on ratings of appearance of distinct dissociative identities, $F(1, 23) = 3.31$, $p = .08$, $\eta_p^2 = .13$, behavior, $F(1, 23) = 3.31$, $p = .08$, $\eta_p^2 = .13$ and facial characteristics, $F(1, 23) = 3.60$, $p = .07$, $\eta_p^2 = .14$, as well as presentation of feigned symptoms, $F(1, 23) = 3.80$, $p = .06$, $\eta_p^2 = .14$. The experimenter ratings suggest that although DID participants and simulators were generally indistinguishable in their presentation of having distinct affect, body posture, and voice characteristics across the identities, DID participants presented as having more distinct behaviors, facial characteristics, and dissociative identities. Simulator participants were judged as presenting with more feigned symptoms.

## Recall

Recall mean scores are reported in Table 3. To test our hypotheses we used ANOVA as this technique is considered robust against violations of assumptions of homogeneity of variance [34] and normality [35,36]. In line with hypothesis 1, the mixed measures ANOVA on Vignette (vignette presented in identity A vs. vignette presented in identity B) by Group (DID, simulator, partial information comparison, full information comparison) indicated a significant Vignette x Group interaction, reflecting a difference in recall between groups across vignettes, $F(3, 63) = 29.98$, $p < .001$, $\eta_p^2 = .59$. Subsequent within group comparisons showed that in line with predictions, DID participants presented a trend to recall significantly less words from the vignette encoded by identity B compared to that experienced by identity A, $t(11) = 1.94$, $p = .09$, whereas a similar but more pronounced effect was found in the simulators, $t(13) = 8.22$, $p < .001$, and partial information comparisons, $t(20) = 15.66$, $p < .001$. As

**Table 3. Vignette-dependent and overall means (with SD) for recall, recognition and quality.**

| | DID (*n* = 12) | Simulators (*n* = 14) | Partial Information Comparison (*n* = 21) | Full Information Comparison (*n* = 20) |
|---|---|---|---|---|
| Vignette-dependent recall | | | | |
| Hit rate Vignette 1 | .20 (.17) [a, b, c] | .39 (.18) | .53 (.15) | .42 (.19) |
| Hit rate Vignette 2 | .08 (.14) [a, b, c] | .00 (.00) | .00 (.00) | .56 (.24) |
| Vignette-dependent recognition | | | | |
| Hit rate Vignette 1 | .67 (.18) | .83 (.18) | .86 (.12) | .84 (.16) |
| Hit rate Vignette 2 | .35 (.18) | .01 (.05) | .02 (.04) | .88 (.12) |
| False alarm rate Vignette 1 | .03 (.05) | .05 (.10) | .03 (.08) | .04 (.06) |
| False alarm rate Vignette 2 | .01 (.03) | .02 (.04) | .02 (.07) | .03 (.05) |
| Sensitivity Vignette 1 | 2.05 (.59) [b] | 2.45 (.62) | 2.65 (.47) | 2.53 (.60) |
| Sensitivity Vignette 2 | 1.07 (1.14) [a, b, c] | -.02 (.19) [e] | .00 (.24) [f] | 2.70 (.51) |
| Response bias Vignette 1 | .51 (.30) [b] | .20 (.38) | .20 (.29) | .22 (.28) |
| Response bias Vignette 2 | 1.05 (.56) [a, b, c] | 1.55 (.16) [e] | 1.55 (.16) [f] | .16 (.20) |
| Quality Remember/Know | | | | |
| Remember Vignette 1 | .46 (.24) | .73 (.29) | .73 (.23) | .72 (.21) |
| Remember Vignette 2 | .17 (.19) [c] | .01 (.03) [e] | .01 (.04) [f] | .84 (.15) |
| Know Vignette 1 | .22 (.17) | .10 (.21) | .13 (.16) | .12 (.15) |
| Know Vignette 2 | .27 (.30) | .01 (.03) | .00 (.02) | .04 (.09 |

Note: *p* < .05

a = DID-Simulator difference

b = DID-partial information comparison difference

c = DID-full information comparison difference

d = Simulator-partial information difference

e = Simulator-full information difference

f = Partial-full information difference.

predicted, a similar effect was absent for the full information comparisons who in fact showed an opposite trend with increased recall for the second vignette, $t(19)$ = -1.81, $p$ = .09.

For Vignette 1, a significant difference was present for Group, $F(3, 63)$ = 9.20, $p$ < .001, $\eta_p^2$ = .31, with Gabriel's post-hoc tests indicating that DID participants had significantly lower recall of information they had experienced in identity A than simulators ($p$ = .04), partial information comparisons ($p$ < .001) and full information comparisons ($p$ = .01). The simulators did not differ significantly from partial ($p$ = .12) and full information comparisons ($p$ = .99), nor did the comparison groups differ ($p$ = .24).

For Vignette 2, the groups also differed, $F(3, 63)$ = 64.42, $p$ < .001, $\eta_p^2$ = .75, with DID participants, simulators and partial information comparisons showing less recall for identity B's vignettes compared with the full information comparisons ($p$ < .001; $p$ < .001; $p$ < .001, respectively). The DID participants did not differ significantly on recall from simulators ($p$ = .61) and partial information comparisons ($p$ = .50), nor did simulators and full information comparisons differ ($p$ = 1.00).

Unrelated to the hypothesis there was also a significant main effect for Vignette, $F(1, 63)$ = 57.23, $p$ < .001, $\eta_p^2$ = .48, indicating that participants generally showed relatively strong recall of stimuli from the first vignette. The Group main effect was also significant, $F(3, 63)$ = 34.86, $p$ < .001, $\eta_p^2$ = .62. Gabriel's post-hoc tests indicated that the recall of DID participants was overall not different from simulators ($p$ = .77), but they did overall recall significantly less information than partial ($p$ = .01) and full information comparisons ($p$ < .001). Simulators

and partial information comparisons did not differ in their recall ($p = .31$), but did recall significantly less stimuli than full information comparisons ($ps < .001$).

## Recognition

Recognition mean scores are shown in Table 3.

**Sensitivity ($d'$).** In line with hypothesis 2, the mixed measures ANOVA on Vignette (presented in identity A vs. in identity B) by Group (DID, simulator, partial information comparison, full information comparison) showed a significant Vignette x Group interaction, reflecting a difference in sensitivity between groups across vignettes, $F(3, 63) = 49.63$, $p < .001$, $\eta_p^2 = .70$. In line with predictions, subsequent within group comparisons indicated that DID participants scored higher on sensitivity for identity A material compared to identity B, $t(11) = 2.62$, $p = .024$. Simulators, $t(13) = 14.53$, $p < .001$, and partial information comparisons, $t(20) = 25.27$, $p < .001$, were also more sensitive in identity A than identity B. Further in line with the pattern of predictions, the full information comparisons did not show a significant difference between both vignettes, $t(19) = -.87$, $p = .40$.

For vignette 1, groups differed significantly on sensitivity, $F(3, 63) = 3.04$, $p = .04$, $\eta_p^2 = .13$, with Gabriel's post hoc tests indicating the DID participants scored significantly lower than partial information comparisons ($p = .02$), although they did not differ significantly from simulators ($p = .35$) or full information comparisons ($p = .12$). Simulators did not differ from partial ($p = .89$) or full information comparisons ($p = .99$), nor did the comparison groups differ ($p = .98$). For vignette 2, the groups differed, $F(3, 63) = 94.40$, $p < .001$, $\eta_p^2 = .82$, with DID participants demonstrating significantly higher sensitivity (i.e., they identified some of identity B's experience as their own) than simulators ($p < .001$) and partial information comparisons, but significantly lower sensitivity than full information comparisons ($p < .001$). Simulators and partial information comparisons demonstrated significantly lower sensitivity than full information comparisons ($p < .001$; $p < .001$, respectively), while simulators and partial information comparisons did not differ significantly ($p = 1.00$). This suggests that DID participants, simulators, and partial information comparisons were less sensitive in discriminating between previously experienced stimuli and distractors for vignette 2, while full information comparisons differentiated vignette 2 from distractor stimuli and showed no difference in sensitivity between identity A and identity B stimuli.

Unrelated to the hypothesis there was also a significant main effect for Vignette, $F(1, 63) = 208.33$, $p < .001$, $\eta_p^2 = .77$, indicating greater sensitivity of vignette 1 compared to vignette 2. The Group main effect was also significant, $F(3, 63) = 49.17$, $p < .001$, $\eta_p^2 = .70$, with the DID, simulator, and partial information comparison groups being significantly less sensitive than the full information comparison group ($ps < .001$) at differentiating between target and distractor stimuli (i.e., whether an item was a target heard in vignette 1 or 2, or whether it was a distractor they were not previously exposed to). The DID participants did not differ from the simulators ($p = .18$) or partial information comparisons ($p = .49$), nor did the simulators from the partial information comparisons ($p = .96$).

**Response bias.** In line with hypothesis 2, the mixed measures ANOVA on Vignette (vignette presented in identiy A vs. vignette presented in identity B) by Group (DID, simulator, partial information comparison, full information comparison) indicated a significant Vignette x Group interaction, reflecting a difference in response bias between groups across vignettes, $F(3, 63) = 61.20$, $p < .001$, $\eta_p^2 = .75$. Subsequent within group comparisons showed that DID participants scored higher on response bias for identity B material compared to identity A, $t(11) = -3.09$, $p = .010$. Simulators, $t(13) = -14.48$, $p < .001$, and partial information comparisons, $t(20) = -27.77$, $p < .001$, also reported more response bias in identity B than in

identity A. Further in line with the pattern of predictions, the full information comparisons did not show a significant difference, $t(19) = .77$, $p = .45$.

For vignette 1, groups differed significantly on response bias, $F(3, 63) = 3.15$, $p = .031$, $\eta_p^2 = .13$, with Gabriel's post hoc tests indicating DID participants scored higher on response bias than partial information comparisons ($p = .04$). They also showed a trend towards significantly more response bias than simulators ($p = .08$) but did not differ from full information comparisons ($p = 0.71$). Simulators did not differ significantly from partial ($p = 1.00$) or full information ($p = 1.00$) comparisons, nor did the comparison groups significantly differ ($p = 1.00$). For vignette 2, the groups differed, $F(3, 63) = 101.97$, p < .001, $\eta_p^2 = .83$, with DID participants, simulators and partial information comparisons scoring significantly higher on response bias than full information comparisons ($p < .001$; $p < .001$; $p < .001$, respectively). Simulators and partial information comparisons were also significantly higher on response bias than DID participants ($p < .001$; $p < .001$, respectively), while simulators and partial information comparisons did not differ significantly ($p = 1.00$). This suggests that simulators and partial information comparisons were biased towards being less likely to indicate that they recognized sentences to represent events from the vignettes at vignette 2. Comparatively, DID participants were more likely to bias towards recognizing sentences from vignette 2, but like the simulator and partial information groups, were less likely to recognize these sentences than the full information comparisons.

Unrelated to the hypothesis there was also a significant main effect for Vignette, $F(1, 63) = 279.20$, $p < .001$, $\eta_p^2 = .82$, indicating greater response bias in vignette 2 compared to vignette 1. The Group main effect was also significant, $F(3, 63) = 38.90$, $p < .001$, $\eta_p^2 = .65$, with the DID, simulator, and partial information comparisons reporting higher response bias than the full information comparison group ($ps < .001$). The DID participants did not differ significantly from the simulators ($p = .87$) or partial information comparisons ($p = .82$), nor did the simulators differ from the partial information comparisons ($p = 1.00$).

## Remember and know responses

Scores indicating the quality of recognition are presented in Table 3. In line with hypothesis 3, the mixed measures ANOVA for Vignette (presented in identity A vs. identity B) by Group (DID, simulator, partial information comparison, full information comparison), for remember responses indicated a significant Vignette x Group interaction, reflecting a difference in remembering between groups across vignettes, $F(3, 63) = 37.19$, $p < .001$, $\eta_p^2 = .64$. In line with predictions, subsequent within group comparisons indicated that DID participants scored higher on remembering for identity A material compared to identity B, $t(11) = 5.37$, $p < .001$, while simulators, $t(13) = 10.97$, $p < .001$, and partial information comparisons, $t(20) = 8.79$, $p < .001$, were also higher on remembering in identity A than identity B. Further in line with the pattern of predictions, the full information comparisons characterized vignette 2 responses more as remember recognitions compared to vignette 1, $t(19) = -3.13$, $p = .01$.

A one-way ANOVA on Group for vignette 1 remember responses, $F(3, 63) = 2.29$, $p = .09$, $\eta_p^2 = .10$, indicated a trend towards differences between groups. No post hoc tests were significant ($p > .13$).

A one-way ANOVA on Group for vignette 2 remember responses showed a significant effect, $F(3, 63) = 53.91$, $p < .001$, $\eta_p^2 = .72$, with DID participants, simulators, and partial information comparisons showing significantly lower remember responses than the full information comparisons ($p < .001$). No other comparisons were significant ($p's > .76$).

Unrelated to the hypothesis, there was also a significant main effect for Vignette, $F(1, 63) = 179.97$, $p < .001$, $\eta_p^2 = .74$, indicating more remember responses for identity A than identity B

vignettes. The Group main effect was also significant, $F(3, 63) = 25.57$, $p < .001$, $\eta_p^2 = .55$, with DID, simulators, and partial information comparisons reporting significantly lower remember scores than full information comparisons ($ps < .001$). The DID participants did not differ from simulators ($p = .99$) or partial information comparisons ($p = .74$), nor did simulators and partial information comparisons differ ($p = .99$).

In contrast to hypothesis 3, for know responses, the Vignette x Group interaction was not significant, $F(3, 63) = 1.38$, $p = .26$, $\eta_p^2 = 0.61$. Unrelated to the hypothesis, the main effect for Vignette was not significant, $F(1, 63) = 1.17$, $p = .17$, $\eta_p^2 = .03$. The Group main effect was significant, $F(3, 63) = 8.38$, $p < .001$, $\eta_p^2 = .29$, with the DID participants having significantly more know responses to stimuli than simulators ($p = .001$), partial information comparisons ($p < .001$) and full information comparisons ($p = .002$). Simulators were not significantly different to partial information comparisons ($p = 1.00$) or full information comparisons ($p = 1.00$), nor were partial or full information comparisons ($p = 1.00$).

## Discussion

The present study assessed the extent of transfer of episodic self-referential memory across amnesic identities in DID. On free recall, DID participants and simulators reported (in the case of DID a trend for) less memory of stimuli presented to identity B compared to identity A, when tested in identity A. Partial information comparisons only recalled stimuli that were presented to them, while full information comparisons showed a trend towards a recency effect (i.e., increased memory for the most recently presented stimuli, so vignette 2/identity B stimuli). On tests of forced choice recognition, results were similar, DID participants and simulators reported less memory of stimuli presented to identity B, when tested in identity A, while partial information comparisons showed no recognition of vignette 2 stimuli, and full information comparisons showed no differences in memory for both sets of stimuli. DID participants, simulators, and partial information comparisons showed a significant reduction in recall and recognition of information presented to identity B, yet the DID group showed more recognition sensitivity and less conservative responses to identity B's experience than the simulators and partial information comparisons, making interpretation somewhat equivocal. This tentatively suggests some stimuli given to identity B were recognized as part of identity A's episodic experience.

On the remember/know paradigm, DID participants and simulators indicated a reduction in remember responses for identity B material compared to identity A material. Partial information comparisons also showed a reduction in remember responses for material that had not been presented to them. Full information comparisons presented no differences in remember responses for vignette stimuli. For know responses, no differences between the identity A and identity B material was found for any of the groups. These results provide support for memories being paired with autonoetic consciousness when retrieved in identity A, indicating they are retrieved from the episodic memory system.

DID participants selected identities for participation that subjectively reported amnesia for material experienced in a different, reported amnesic, identity. The report of subjective amnesia was checked during testing by the experimenter by asking the patients after a switch whether indeed they did not remember what was done in the other identity, which patients verified. On the measures of memory retrieval, the results suggested evidence of inter-identity amnesia. DID participants showed a trend for amnesia on tests of recall and similarly reported less recognition for stimuli from the vignette they had experienced in the other identity compared to that experienced in the same identity.

## Study 2

Extending the findings of study 1, this study aimed to determine whether the reported impairment in memory transfer across partial information identities was evident for episodic autobiographical memories (i.e., information from actually experienced events). Episodic autobiographical memory has not been well assessed in inter-identity studies in DID, but such memories are at the core of diagnostic criteria and general phenomenology of DID, where patients report amnesia for experiences they have had [e.g., 37]. To assess the pervasiveness of amnesia in DID, the three comparison samples used in study 1 were included in this study as well.

For DID participants, two identities were used who reported amnesia for events experienced by the other. As in study 1, free recall and forced choice recognition tasks were used to assess explicit memory retrieval for events experienced in their own identity compared with the other. The remember/know paradigm was included to assess whether, in the presence of transfer, the memories were accessed with noetic or autonoetic consciousness [21]. The current study tested the following three hypotheses: (1) For free recall, DID participants would exhibit amnesia for behavioral tasks encoded in a different amnesic identity, showing a similar profile to simulator and partial information comparison recall performance; (2) For forced choice recognition, a similar pattern of results was predicted; (3) For remember and know responses, DID participants were expected to exhibit a higher level of autonoetic consciousness for memories encoded in the same identity compared to those encoded in the other identity, comparable to simulator and partial information comparisons. Full information comparisons should exhibit comparable levels of autonoetic and noetic performance across the different materials.

## Method

### Participants

**DID sample.** Recruitment, inclusion and exclusion criteria were the same as for study 1, with the same participants assessed. As a result of the inclusion and exclusion criteria, five participants were removed from the recruitment total of 19, for being unable to complete the task ($n = 1$), being unable to switch between identities ($n = 3$), or experiencing test interference from a third identity ($n = 1$). Fourteen participants were included in the final sample. Of the 14 DID participants tested, one was removed from the free recall analysis for failing to complete the task.

**Comparison samples.** The same participants, recruitment, inclusion and exclusion criteria were used as in study 1.

**Simulator sample.** Participants, recruitment, inclusion and exclusion criteria were the same as for study 1. Two simulators who did not complete the task were removed from the recall analysis.

Table 4 displays the demographic details for each sample. For the sample included in the free recall task, the groups presented a significant difference on age, $F(3, 64) = 3.81$, $p = .014$, $\eta_p^2 = .15$, with simulators significantly younger than DID participants ($p = .03$) and partial information comparisons ($p = .04$), and trending towards significance when compared with full information comparisons ($p = .07$). No other comparisons were significant. For the samples included in the forced choice recognition task, the groups presented a trend to differ significantly for age, $F(3, 67) = 2.48$, $p = .07$, $\eta_p^2 = .10$. For both tasks, there were minimal variations in gender. Inferential statistics were also not utilised for ethnicity and education due to cells with low counts.

**Table 4. Participant demographic data across groups.**

| | DID Free Recall Task (n = 13) | DID Forced Choice Task (n = 14) | Simulator Free Recall Task (n = 14) | Simulator Forced Choice Task (n = 16) | Partial Information Comparison (n = 21) | Full Information Comparison (n = 20) |
|---|---|---|---|---|---|---|
| Age M (SD) | 41.00 (3.45) | 41.29 (3.21) | 28.86 (3.85) | 31.38 (3.79) | 39.10 (1.90) | 38.35 (1.62) |
| Gender n(%) | | | | | | |
| Male | 0 (0%) | 0 (0%) | 2 (14.3%) | 2 (12.5%) | 1 (4.8%) | 1 (5%) |
| Female | 13 (100%) | 14 (100%) | 12 (85.7%) | 14 (87.5%) | 20 (95.2%) | 19 (95%) |
| Ethnicity* | | | | | | |
| New Zealand European | 0 (0%) | 0 (0%) | 12 (85.7%) | 13 (81.3%) | 11 (52.4%) | 16 (80%) |
| Maori | 0 (0%) | 0 (0%) | 0 (0%) | 1 (6.3%) | 0 (0%) | 0 (0%) |
| Australian European | 10 (76.9%) | 11 (78.6%) | 2 (14.3%) | 2 (12.5%) | 1 (4.8%) | 0 (0%) |
| European Other | 3 (23.1%) | 3 (21.4%) | 1 (7.1%) | 2 (12.5%) | 5 (23.8%) | 0 (0%) |
| Other | 2 (15.4%) | 2 (14.3%) | 0 (0%) | 0 (0%) | 5 (23.8%) | 4 (20%) |
| Qualification§ | | | | | | |
| High school certificate | 3 (23.1%) | 4 (28.6%) | 7 (50%) | 7 (43.8%) | 2 (9.5%) | 2 (10%) |
| Post-high school non-university (e.g. Trade certificate) | 2 (15.4%) | 2 (14.3%) | 0 (0%) | 0 (0%) | 8 (38.1%) | 7 (35%) |
| University level | 6 (46.2%) | 6 (42.9%) | 7 (50%) | 9 (56.2%) | 11 (52.4%) | 11 (55%) |

* For the DID groups, two participants chose two ethnicities.

For the free recall simulator group, one participant chose two ethnicities. For the forced choice recognition simulator group, two participants chose two ethnicities. For the amnesic comparison group, one participant chose two ethnicities. For the nonamnesic comparison group, one participant chose two ethnicities, and one participant did not indicate an ethnicity.

§For the DID groups, two participants reported attaining no qualifications.

## Materials

All participants completed the behavioral tasks and a questionnaire battery.

**Behavioral tasks.** Two sets of behavioral tasks were created for the study and were designed to provide unique episodic autobiographical experiences that could be used to assess memory retrieval across the two identity states that subjectively reported amnesia. The first set of tasks required participants to take out their mobile phone, place it on silent and put it by a pot plant. They were then instructed to find a book which was placed under a stack of white papers. They retrieved a pamphlet from inside the book and were instructed to find a loose piece of paper which had pictures of a circle, triangle, and square printed on it. Under each shape were instructions on how to draw them. Participants were then required to draw the shapes on a blank piece of paper in the same orientation printed on the original sheet. They coloured the circle in blue, the triangle in red, and the square in green using coloured pencils. Participants were then asked to choose their favourite coloured pen and write their first name under the circle and using their second favourite coloured pen, their birthday under the square. Participants were then asked to look at their drawn shapes and think of a memory that it reminded them of. They were asked to visualise this memory for 15 seconds.

The second set of tasks involved participants tracing a picture of a dog followed by filling up three plastic cups with water to three different levels depicted by lines drawn on the side of the cups. They were then asked to place the cups in order from most to least filled. Participants then completed an origami task where they followed the instructions of the experimenter to create a finger pointer. Participants then used this to point to five photos of celebrities and name them. They were asked to put them in order from their most to least liked, choose the

celebrity they saw as most similar to themselves and provide a reason for their decision. The complete instructions for each behavioral task can be found at https://osf.io/u4r6k/.

**Questionnaires.** All participants completed the same questionnaires as assessed in study 1, and the alphas of the DES and PSS-SR were satisfactory across all groups ($\alpha > .77$).

## Experimental measures

**Free recall (FR).** To assess episodic autobiographical experience in a more natural way, there was no immediate recall task utilized after each behavioral exercise. Delayed free recall followed the same procedure as in Study 1, but the time between concluding the behavioral exercises and completing the recall task was approximately 10 minutes. Following their recall of one of the sets of tasks, if they did not freely recall the other tasks (i.e., DID participants, simulators), they were prompted with, "did you take part in any other tasks?" A positive response brought an invitation to detail those tasks. A negative response resulted in the researcher presenting an additional prompt of "do you remember anything about a mobile phone [or tracing task]?" If the response was negative, the researcher asked, "do you remember anything about drawing shapes [or filling cups]?" The prompts were included to assess if DID participants and simulators could retrieve information of events experienced in their amnesic identity. Partial information comparisons were also prompted to recall a second set of behavioral tasks, despite only being exposed to one set.

**Forced choice recognition.** *Stimuli Recognition*. Participants were presented with 40 sentences each individually placed in the centre of a computer screen. Half of these were related to each Behavioral task (10 for each task; e.g., Did you colour in the shapes?), and 20 were not related to either set of behavioral tasks (distractor stimuli; e.g., Did you paint over the lines?). Participants specified whether they recognized the sentences as representing details of events from the Behavioral task/s they experienced by pressing the Y key on the keyboard, or if they did not recognize them, by pressing the N key. The order of sentence presentation was randomized across each participant. The forced choice recognition task was administered 20 minutes after exposure to the second and final set of behavioral tasks. During this time interval, participants performed unrelated tasks.

*Remember/know*. Equivalent to study 1, if participants recognized a sentence as representing an experience from the Behavioral tasks, they were asked to establish whether they 'remembered' or 'knew' that it had happened.

## Procedure

The procedure of this study follows that used in study 1, with transfer of the behavioral task stimuli being assessed.

## Data analysis

Data analysis of recall, recognition and remember/know tasks follows that used in study 1. Data for study 2 can be found at: https://osf.io/u4r6k/

## Results

### Questionnaires

DES and PSS-SR scores across groups were largely the same as study 1 (see Table 1, https://osf.io/u4r6k/).

## Post-experiment simulator performance questionnaire

Rating for DID presentation as assessed by the researcher who was blind to group classification for DID and simulators, was generally similar to study 1. These ratings suggested that DID participants and simulators were indistinguishable on items associated with their presentation of amnesia and appearance of their identities, however the simulators were significantly less likely to be rated as presenting with genuine dissociative identities, $F(1, 26) = 4.39$, $p = .046$, $\eta_p^2 = .14$, and were more likely to be rated as feigning their symptoms, $F(1, 26) = 4.32$, $p = .048$, $\eta_p^2 = .14$.

## Recall

Recall mean scores are reported in Table 5. In line with hypothesis 1, the mixed ANOVA on Behavioral task (Behavioral task 1 presented first in identity A, Behavioral task 2 presented second in identity B) by Group (DID, simulator, partial information comparison, full information comparison) indicated a significant Task x Group interaction, reflecting a difference in recall between groups across tasks, $F(3, 64) = 28.72$, $p < .001$, $\eta_p^2 = .57$. Subsequent within group comparisons showed that in line with predictions, DID participants presented a trend to recall significantly less experiences from tasks encoded by identity B compared to that experienced by identity A, $t(12) = 2.10$, $p = .057$, whereas a similar but more pronounced effect was found in simulators, $t(13) = 6.76$, $p < .001$, and partial information comparisons, $t(20) = 8.24$, $p <$

**Table 5. Time-dependent and overall means (with SD) for recall, recognition and quality.**

| | DID (*n* = 13/14) | Simulators (*n* = 14/16) | Partial Information Comparison (*n* = 21) | Full Information Comparison (*n* = 20) |
|---|---|---|---|---|
| Behavioral Task-dependent recall | | | | |
| Hit rate task 1 | .29 (.18) [b, c] | .42 (.22) [d, e] | .67 (.26) | .68 (.16) |
| Hit rate task 2 | .13 (.20) [c] | .00 (.01) [e] | .03 (.14) [f] | .77 (.12) |
| Behavioral Task-dependent recognition | | | | |
| Hit rate task 1 | .79 (.16) | .92 (.12) | .89 (.23) | .93 (.09) |
| Hit rate task 2 | .38 (.36) | .03 (.05) | .10 (15) | .95 (.06) |
| False alarm rate task 1 | .04 (.09) | .02 (.05) | .03 (.06) | .02 (.05) |
| False alarm rate task 2 | .03 (.06) | .03 (.09) | .02 (.06) | .02 (.04) |
| Sensitivity task 1 | 2.25 (.52) [a, b, c] | 2.73 (.42) | 2.74 (.38) | 2.74 (.42) |
| Sensitivity task 2 | 1.06 (1.03) [a, b, c] | .00 (.25) [e] | .27 (.45) [f] | 2.82 (.27) |
| Response bias task 1 | .27 (.34) | .12 (.22) | .06 (.23) | .10 (18) |
| Response bias task 2 | .90 (.60) [a, b, c] | 1.43 (.19) [e] | 1.34 (.29) [f] | .06 (.15) |
| Quality Remember/Know | | | | |
| Remember task 1 | .76 (.23) [a] | .96 (.08) | .96 (.13) | .93 (.14) |
| Remember task 2 | .12 (.16) | .01 (.04) [e] | .07 (.15) | .54 (.43) |
| Know task 1 | .24 (.23) | .04 (.08) | .04 (.13) | .07 (.14) |
| Know task 2 | .13 (.14) | .01 (.03) | .04 (.02) | .08 (.16) |

Note: $p < .05$

a = DID-Simulator difference

b = DID-partial information comparison difference

c = DID-full information comparison difference

d = Simulator-partial information difference

e = Simulator-full information difference

f = Partial-full information difference.

.001. As predicted, full information comparisons failed to show this pattern, and even showed a significant increase in recall for identity B's tasks, $t(19) = -3.10$, $p = .01$, apparently reflecting a recency recall effect.

For Behavioral task 1 (i.e., identity A for DID and simulators), a significant difference was present for group, $F(3, 64) = 13.11$, $p < .001$, $\eta_p^2 = .38$, with DID participants having significantly lower recall of tasks they completed in identity A than partial and full information comparisons ($p < .001$). Simulators also differed from comparison groups ($p = .01$; $p = .01$), but not from the DID participants ($p = .49$). The comparison groups did not differ ($p = 1.00$).

For Behavioral task 2 (i.e., identity B for DID and simulators), the groups also differed, $F(3, 64) = 149.45$, $p < .001$, $\eta_p^2 = .88$, with DID participants, simulators and partial information comparisons showing less recall for identity B's tasks compared to the full information comparisons ($p < .001$; $p < .001$; $p < .001$, respectively). DID participants trended towards recalling more tasks from the events experienced in Identity B than simulators ($p = .08$). DID participants did not differ significantly on recall compared with partial information comparisons ($p = .16$), nor did simulators and partial information comparisons ($p = .99$).

Unrelated to the hypothesis, there was a main effect for Behavioral task, $F(1, 64) = 72.69$, $p < .001$, $\eta_p^2 = .53$, indicating more recall of stimuli from the first Behavioral task. The Group main effect was also significant, $F(3, 64) = 78.78$, $p < .001$, $\eta_p^2 = .79$. Gabriel's post hoc tests indicated that recall for DID participants was overall not different to simulators ($p = 1.00$), but they did recall significantly fewer tasks than partial (who only got identity A stimuli; $p = .01$) and full information comparison groups ($p < .001$). Simulators also recalled significantly less stimuli than partial ($p = .01$) and full information comparison groups ($p < .001$), and partial information comparisons recalled significantly less stimuli than full information comparison groups ($p < .001$).

## Recognition

Recognition memory scores are shown in Table 5. One stimulus from the mobile phone behavioral task set and two stimuli from the tracing dog behavioral task set were removed from the recognition sentences due to being rated as too similar to other stimuli. These omissions were to ensure all stimuli were unique, to most thoroughly assess inter-identity amnesia for unrelated material.

**Sensitivity ($d'$).**   In line with hypothesis 2, the mixed measures ANOVA on Behavioral task (presented in identity A vs. in identity B) by Group (DID, simulator, partial information comparison, full information comparison) indicated a significant Behavioral task x Group interaction, reflecting a difference in sensitivity between groups across Behavioral tasks $F(3, 67) = 71.76$, $p < .001$, $\eta_p^2 = .76$. In line with predictions, subsequent within group comparisons indicated that DID participants were more sensitive to recognize stimuli for identity A material compared to identity B, $t(13) = 3.84$, $p = .002$. Simulators, $t(15) = 27.56$, $p < .001$, and partial information comparisons, $t(20) = 20.28$, $p < .001$, were also more sensitive in identity A than identity B. However, further in line with the pattern of predictions, the full information comparisons did not show a significant difference between behavioral tasks, $t(19) = -.88$, $p = .388$.

For Behavioral task 1, groups differed significantly on sensitivity, $F(3, 67) = 4.73$, $p = .005$, $\eta_p^2 = .175$, with DID participants significantly lower than simulators ($p = .02$), partial information comparisons ($p = .01$) and full information comparisons ($p = .01$), suggesting DID participants exhibit a poorer ability to discriminate between target and distractor stimuli than simulators and non-clinical comparison groups. Simulators did not differ from partial information comparisons ($p = 1.00$) or full information comparisons ($p = 1.00$), nor did the

comparison groups differ ($p$ = 1.00). For Behavioral task 2, the groups differed, $F$(3, 67) = 31.17, $p$ < .001, $\eta_p^2$ = .82, with DID participants significantly higher on sensitivity than simulators ($p$ < .001) and partial information comparisons ($p$ < .001), but significantly lower on sensitivity than full information comparisons ($p$ < .001). Simulators and partial information comparisons demonstrated significantly lower sensitivity than full information comparisons ($p$ < .001; $p$ < .001, respectively), but did not differ from each other ($p$ = .59). These results suggest that while DID participants recognized more experiences had as identity B when tested in identity A compared to simulators and partial information comparisons (i.e., the DID sample had higer sensitivity scores), the three groups (DID, simulators, partial information comparisons) were better at detecting identity A (Task 1) compared to identity B (Task 2) experiences. Full information comparisons showed no difference in sensitivity between Behavioral tasks 1 and 2.

Unrelated to the hypothesis, there was also a significant main effect for Behavioral Task, $F$(1, 67) = 388.35, $p$ < .001, $\eta_p^2$ = .85, indicating a greater sensitivity of Behavioral task 1 compared to Behavioral task 2. That is, participants were better able to discriminate between target and distractor stimuli from Behavioral task 1 than Behavioral task 2. The Group main effect was also significant, $F$(3, 67) = 60.95, $p$ < .001, $\eta_p^2$ = .73, with DID, simulator and partial information comparison groups being significantly less sensitive than the full information comparison group at recognising stimuli from identity A and B's experiences than distractor stimuli ($ps$ < .001). The DID participants did not differ from simulators ($p$ = .17) or partial information comparisons ($p$ = .78), nor did the simulators differ from the partial information comparisons ($p$ = .80).

**Response bias.** In line with hypothesis 2, the mixed measures ANOVA on Behavioral task (presented in identity A vs. identity B) by Group (DID, simulator, partial information comparison, full information comparison) indicated a significant Behavioral task x Group interaction, $F$(3, 67) = 52.78, $p$ < .001, $\eta_p^2$ = .70. In line with predictions, subsequent within group comparisons indicated that DID participants reported a higher response bias in identity B than identity A, $t$(13) = -3.37, $p$ = .01. Simulators, $t$(15) = -21.03, $p$ < .001, and partial information comparisons, $t$(20) = -20.11, $p$ < .001, also reported more response bias in identity B than identity A. Further in line with the pattern of predictions, the full information comparisons did not show a significant difference between behavioral tasks, $t$(19) = .75, $p$ = .46.

For Behavioral task 1, groups presented a trend to differ significantly on response bias, $F$(3, 67) = 2.24, $p$ = .09, $\eta_p^2$ = .09. For Behavioral task 2, the groups differed, $F$(3, 67) = 70.33, $p$ < .001, $\eta_p^2$ = .76, with DID participants, simulators and partial information comparisons scoring significantly higher on response bias than full information comparisons ($p$ < .001; $p$ < .001; $p$ < .001, respectively). Simulators and partial information comparisons were also significantly higher on response bias than DID participants ($p$ < .001; $p$ = .001, respectively), while simulators and partial information comparisons did not differ significantly ($p$ = .94). This suggests that DID participants were less likely to indicate they recognized sentences that represent events from identity B's behavioral tasks (i.e., more conservative) than full information comparisons, but more likely (i.e., less conservative) than simulators and partial information comparisons.

Unrelated to the hypothesis, there was also a significant main effect for Behavioral task, $F$(1, 67) = 293.50, $p$ < .001, $\eta_p^2$ = .81, with greater response bias (i.e., participants were more conservative so less inclined to recognize items) in Behavioral task 2 compared to Behavioral task 1. The Group main effect was also significant, $F$(3, 67) = 41.02, $p$ < .001, $\eta_p^2$ = .65, with DID, simulator and partial information comparison groups reporting higher response bias than full information comparisons ($ps$ < .001).

## Remember and know responses

The mixed ANOVA for Behavioral task by Group for remember responses indicated a significant effect, $F(3, 67) = 3.66$, $p = .02$, $\eta_p^2 = .14$. Subsequent within group comparisons indicated that DID participants, $t(13) = 4.17$, $p = .001$, simulators, $t(15) = 8.60$, $p < .001$, partial information comparisons, $t(20) = 6.58$, $p < .001$, and full information comparisons, $t(19) = 3.32$, $p = .004$, characterized Behavioral task 1 responses more as remember recognitions compared to Behavioral task 2 (n.b., partial information comparisons did not receive a second set of behavioral tasks).

A significant one-way ANOVA on Group for Behavioral task 1 remember responses, $F(3, 67) = 3.31$, $p = .03$, $\eta_p^2 = .13$, indicated that the DID participants had significantly lower rates of remembering events they experienced in Identity A than simulators ($p = .03$). A trend presented for lower rates of remembering in the DID sample than full information comparisons ($p = .07$). They were not significantly different to partial information comparisons ($p < .13$). Simulators did not differ compared with partial information comparisons ($p = .96$) and full information comparisons ($p = .99$), nor did the comparison groups ($p = 1.00$).

A one-way ANOVA for Group for Behavioral task 2 remember responses also showed a significant effect, $F(3, 67) = 4.04$, $p = .011$, $\eta_p^2 = .15$, with simulators reporting lower remember responses than full information comparisons ($p = .01$). Partial information comparisons reported a trend to report less remember responses than full information comparisons ($p = 0.67$). No other comparisons were significant.

Unrelated to the hypotheses, there was also a significant main effect for Behavioral task, $F(1, 67) = 117.76$, $p < .001$, $\eta_p^2 = .64$, indicating more remember responses for Identity A tasks than Identity B tasks (see Table 5). The Group main effect was also significant, $F(3, 67) = 4.19$, $p = .01$, $\eta_p^2 = .16$, with DID and simulator groups reporting significantly lower remember scores than full information comparisons ($p < .02$; $p < .04$, respectively). Partial information comparisons showed a trend for lower remember scores than full information comparisons ($p = 0.78$). DID participants did not differ from simulators ($p = .99$) or partial information comparisons ($p = .96$), nor did simulators and partial information comparisons ($p = .99$).

For know responses, in contrast to hypothesis 3, the Behavioral task x Group interaction was not significant, $F(3, 67) = 1.17$, $p = .33$, $\eta_p^2 = .05$, reflecting that the groups showed similar levels of 'know' responses across the two behavioral tasks.

The main effect for Behavioral task trended towards significance, $F(1, 67) = 3.00$, $p = .09$, $\eta_p^2 = .04$. The group main effect was significant, $F(3, 67) = 7.33$, $p < .001$, $\eta_p^2 = .25$, with DID participants reporting significantly more know responses across sessions than simulators ($p = .001$), partial information comparisons ($p < .001$), and full information comparisons ($p = .003$). Simulators were not different to partial information comparisons ($p = 1.00$) or full information comparisons ($p = .99$), nor were partial and full information comparisons ($p = .995$).

## Discussion

Study 2 assessed the extent of transfer of episodic autobiographical memory across amnesic identities in DID. Subjectively, DID participants reported amnesia for events experienced in their other, amnesic identity. On tests of free recall, DID participants showed a trend to report less sentences related to stimuli presented to identity B, when tested in identity A ($p = .057$). Simulators and partial information comparisons similarly showed a reduction in recall of identity B material, while full information comparisons showed the opposite pattern reflecting an increase in memory of the most recently presented stimuli (i.e., Behavioral task 2/identity B stimuli; a recency effect). On tests of forced choice recognition, results were similar. DID participants and simulators reported less memory of stimuli presented to identity B, when tested in

identity A, while partial information comparisons unsurprisingly showed less recognition of Behavioral task 2 stimuli and full information comparisons showed no difference in remembering either set of stimuli. DID participants, simulators and partial information comparisons showed greater response bias when responding to identity B tasks (i.e., were more likely to indicate they had not experienced the events outlined), while full information comparisons showed no difference in response bias or sensitivity between both sets of stimuli. On the remember/know paradigm, all groups indicated a reduction in remember responses for identity B material (i.e., partial information comparisons also showed a reduction in remember responses for material that had not been presented to them, as expected). For all participants, no differences in know responses for identity A and B material were found. Overall, these results suggest that based on subjective reports of recall and recognition, people with DID show a profile of inter-identity amnesia for episodic autobiographical memory.

However, whereas for identity A, DID participants recognized significantly less stimuli compared to the other groups, in identity B they recognized significantly more stimuli than simulators and partial information comparisons, suggesting a different pattern of responding between these groups and an indication that the reported amnesia in DID was not complete (i.e., some experiences from identity B were recognized). Although the DID participants showed a reduction in recognition of the experiences had by the amnesic identity, they did not present as great an impairment as seen in groups feigning amnesia or exhibiting no familiarity with the stimuli (partial information group). There was also a reduction in remember responses in DID for stimuli experienced in the amnesic identity (Identity B) compared to stimuli experienced in their own identity (Identity A). In contrast, Huntjens et al. [14] showed no differences in the number of remember or know responses from identity A and identity B in their assessment of inter-identity transfer of episodic, non-autobiographical memory. The DID sample reported significantly more know responses to stimuli overall (i.e., for identity A and B stimuli) indicating their memories were characterized relatively more by noetic consciousness.

## General discussion

Although previous research has explored inter-identity amnesia, there has been a lack of studies assessing the transfer of episodic and autobiographical material, with these types of memories greatly associated with the development of sense of self and central to DID clinical phenomenology [37]. In the present studies, DID participants scored significantly lower on recognition sensitivity and showed a more conservative response bias for episodic self-referential and autobiographical stimuli encoded in another identity compared to stimuli encoded in the same identity. The performance of DID participants was largely comparable to members of the general population who were exposed to only one set of stimuli (partial information comparisons). The DID group performance on recall tasks trended towards a profile of reported amnesia in both self-referential and autobiographical tasks. Moreover, DID participants reported a qualitatively different way of retrieving experiences of the other identity, with these events less likely to be retrieved with autonoetic consciousness, whereas no identity differences were found for retrieval with noetic consciousness.

However, simulation of amnesia proved possible on the tasks used in this study, with simulators presenting the profile of amnesia seen in DID participants. Simulators also presented with the same reduction in autonoetic consciousness for other identity material. Due to the parallels between results for DID and simulator groups, it is not possible to determine whether DID participants presented a true inability to access information experienced in another identity, were simulating inter-identity amnesia, or reported amnesia for other reasons. Yet, the

simulation interpretation is inconsistent with results from identity B retrieval data, where compared to the simulator group, DID participants showed more recognition of identity B stimuli (i.e., some leakage into identity A of identity B's experience) for self-referential and autobiographical experiences, and trended in that direction for recall of autobiographical experience. Such findings are inconsistent with feigning amnesia (as shown by simulators), and suggest that despite the DID participants self-report of amnesia, and this being generally consistent with the profile of results, some memory representations may be available for retrieval in so-called 'amnesic' identities. Further work is needed in this area, as more indirect forms of testing explicit retrieval for autobiographical experiences have indicated transfer of episodic autobiographical memories [19]. Further, regarding comparisons with the simulator group, the DID sample not only showed differences in identity B recognition profiles, they also showed higher scores of dissociation and PTSD. In addition, they indicated that they interpreted experimental stimuli as being significantly more negative emotionally. In terms of their presentation, DID participants were more likely to be rated as presenting with changes in appearance based on affect, body posture and voice and facial characteristics as well as the appearance of distinct dissociative identities. These results collectively suggest that DID is not easy to accurately feign, at least not by the current sample of professional and amateur actors. More specific samples (e.g., a selection of individuals high in fantasy proneness), might be worthy of investigation in future studies.

Regarding memory representations being available in self-reported amnesic identities, Huntjens et al. [12,38] argue that a meta-memory problem may explain the subjective reports of inter-identity amnesia. Amnesic identities are said to believe at a metacognitive level that they are unable to retrieve information of events experienced by another identity, and as a result DID participants may not actively search for memories they believe they cannot access (i.e., were encoded in another identity state) or attribute retrieved memories to the other identity state [12]. The theory of a meta-memory impairment suggests that people with DID may not realise the full extent of their memory abilities and the belief in compartmentalization of identities held by those with DID may generate the results of inter-identity amnesia instead of conscious simulation.

The question remains as to how and when identity-dependent dissociative beliefs interfere with the retrieval of memories from other identities. The current understanding of the subjective experience of inter-identity amnesia is that beliefs may either interfere with 1) initiating a search for a specific memory and/or 2) the acknowledgment of memory ownership once a memory is retrieved. Intentional, conscious memory retrieval is reliant on active searches of the autobiographical knowledge base in order to find experiences consistent with current personal goals [39]. Before search actions are engaged, evaluative processes are thought to operate to determine if a search should be initiated or terminated. It is this process which may be hampered by dissociative beliefs. Conway [40] proposed that memory retrieval, and whether search actions are initiated or terminated, is largely determined by how a person perceives the self. In DID the processes described above are hypothesized to be identity-dependent, meaning active searching is limited to memories consistent with the identity's sense of self. Thus no memories outside the identity's purview will be pursued and therefore found, and a subjective sense of amnesia will result.

Whereas the results of the current studies are consistent with this theory, further evidence for the role of dissociative beliefs in subjective retrieval deficits would be to test these beliefs in a sample of patients who report inter-identity amnesia. Moreover, future studies should compare correlations between dissociative beliefs and subjective versus objective memory functioning. Whereas the model presented would predict associations between dissociative beliefs and subjective reports of amnesia assessed using memory tasks that entail intentional memory

search (e.g., free recall tasks), the model would also predict a lack of association between beliefs and more objective, nonintentional memory performance (e.g., implicit retrieval tasks).

A limitation of the study was the partial information comparisons may have found the task easier due to them engaging less cognitive resources as a result of receiving only half of the stimuli. However, as partial information comparisons did not report significantly improved memory performance for their stimuli compared to the simulators and full information comparisons, this limitation does not fully explain their memory profile for test session 1. It is important to recognize the relatively small sample sizes, especially for the DID and simulator groups. Future studies should aim to include larger DID samples to replicate the current findings in a more powerful design. In addition, the current studies present results from a relatively stable DID sample due to the requirements of participants being able to switch on demand and only a small number of simulator participants were administered the DDIS. In addition, because in study 2 the simulator group was significantly younger than the DID group for recall tasks, it is difficult to generalise these findings.

## Conclusions

In summary, DID participants demonstrated a pattern of amnesia across identities that claimed no knowledge of events that occur in the other identity on tests of free recall and forced choice recognition; a pattern similar to those with true amnesia (partial information group) as well as simulators. However, on tests of forced choice recognition the DID sample recognized more stimuli learned in the other (reportedly amnesic) identity (identity B) than either simulators and comparisons who had not encoded the information. As such, DID participants appear to have a somewhat different profile of retrieval to those who had not encoded the information and those presenting with feigned amnesia, and seem to have access to some information that they believe they have amnesia for. Clinicians may seek to use these findings to further understand the extent of DID patients' memory abilities by conceptualising amnesia as a failure to initiate the search process for memories encoded in other identities. Considering inter-identity amnesia more as a meta-memory problem rather than a retrieval inability problem, where the person has access to at least some memory representations they believe they do not, may provide a therapy framework to aid people with DID to integrate their memories across identities.

## Acknowledgments

The authors would like to acknowledge the assistance of Dr. Greta Bond and Lenaire Seager in the development of this work, and the Cannan Institute for supporting data collection.

## Author Contributions

**Conceptualization:** Martin J. Dorahy, Rafaele Huntjens.

**Data curation:** Rosemary J. Marsh, Martin J. Dorahy, Rafaele Huntjens.

**Formal analysis:** Rosemary J. Marsh, Martin J. Dorahy, Rafaele Huntjens.

**Investigation:** Rosemary J. Marsh, Warwick Middleton.

**Methodology:** Martin J. Dorahy, Chandele Butler, Warwick Middleton, Rafaele Huntjens.

**Project administration:** Martin J. Dorahy.

**Supervision:** Martin J. Dorahy, Simon Kemp, Rafaele Huntjens.

**Writing – original draft:** Rosemary J. Marsh.

**Writing – review & editing:** Rosemary J. Marsh, Martin J. Dorahy, Warwick Middleton, Peter J. de Jong, Simon Kemp, Rafaele Huntjens.

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
