## [Decision Letter · Decision Letter 0]

13 Nov 2020

PONE-D-20-30695

Inter-Identity amnesia for neutral episodic self-referential and autobiographical memory in Dissociative Identity Disorder: An assessment of recall and recognition

PLOS ONE

Dear Dr. Dorahy,

Thank you for submitting your manuscript to PLOS ONE. After careful consideration, we feel that it has merit but does not fully meet PLOS ONE’s publication criteria as it currently stands. Therefore, we invite you to submit a revised version of the manuscript that addresses the points raised during the review process.

We look forward to receiving your revised manuscript.

Kind regards,

Vedat Sar, M.D.

Academic Editor

PLOS ONE

Journal Requirements:

2) We note that you have stated that you will provide repository information for your data at acceptance. Should your manuscript be accepted for publication, we will hold it until you provide the relevant accession numbers or DOIs necessary to access your data. If you wish to make changes to your Data Availability statement, please describe these changes in your cover letter and we will update your Data Availability statement to reflect the information you provide.

Reviewers' comments:

Reviewer's Responses to Questions

**Comments to the Author**

1. Is the manuscript technically sound, and do the data support the conclusions?

Reviewer #1: Yes

Reviewer #2: Yes

2. Has the statistical analysis been performed appropriately and rigorously? 

Reviewer #1: Yes

Reviewer #2: Yes

3. Have the authors made all data underlying the findings in their manuscript fully available?

Reviewer #1: Yes

Reviewer #2: Yes

4. Is the manuscript presented in an intelligible fashion and written in standard English?

Reviewer #1: Yes

Reviewer #2: Yes

5. Review Comments to the Author

Reviewer #1: I think that the conceptual framework and methodology are excellent and the findings provide important insights in understanding inter-identity amnesia in this well written paper.

I believe there are only minor revisions required, and often to improve clarity and readability given the complexity of the topic.

The biggest problem I have is with the terminology “amnestic comparisons”, or simply “amnestic” (as compared to non-amnestic) in the abstract. The term needs to be clarified to highlight that these people are not amnestic, and that the process by which they have restricted access to information is not amnesia. Alternative wording – which I will leave up to the authors to suggest is needed along the lines of “amnestic simulators” or “partial information group”. Even identities in this context is confusing, and terms such as memory deficits given there was no memory.

Page 5 – please clarify sentence as: “4) non-clinical comparison participants given stimuli shown to both identities (i.e., all of the stimuli; nonamnesic group).”

Page 5 – reference to “distractor performance” needs more context, or left out until the discussion that describes this.

Page 6 – footnote 1. is important and should be moved to main text, and further clarification. Were these trauma-knowing identifies assigned to A or B, or was there an attempt to evenly distribute them?

Page 8 – perhaps specify that Post-experiment Questionnarie determination of genuine vs feigned assessed all facets of the research, i.e. measures, experiments, overall conduct.

Page 10 - In regards to prompts, how is the number or prompts taken measured/accounted for?

Page 11 – can you please provide more information on know / remember – is each dimension assessed or is it either/or? Is it a Likert scale? What if a respondent doesn’t have a clue if they know/remember the information of think it’s a bit of both?

Page 13 – I think the findings that the similarities between those with DID and simulators is where the information and coaching is, but the overall quality of the dissociative/amnestic experience being different needs to be highlighted in the discussion drawing on a range of findings in this paper, starting here. The paper really show that DID is a complex phenomenon that is incredibly difficult to feign. From their lower DES (I would expect simulators to score similarly high on the DES amnesia subscale, and depersonalization items that touch on being disconnected from the self, but far lower on absorption items). Differences in PTSD symptoms, differences in know/remember, memory being more porous in the DID samples, less likely to say for sure that ‘distractor’ events did not happen. I think there is important findings for the fantasy/trauma debate that need to be highlighted.

Page 21 When we have p = 1.00 can we say the groups were identical, rather than not significantly different?

Page 21 – I am a little uncomfortable with the terms that imply “more liberal”, “more hesistant” “more conservative”. Is that really what is happening? I doubt we know. And if that’s the case then perhaps a “less likely”.

Page 26 – I am uncomfortable with the term “failing to complete”. If one of the participants saw this, it would seem a harsh judgement. Perhaps “unable to complete”.

Page 29 – Can we have exact data on “less likely to be rated as presenting with genuine dissociative identity and more likely to be feigning. Was this statistically significant?

There is an issue with headings. We have discussions, then general discussion, and no conclusion.

Page 37 – Line 7, sentence commencing “Their” – Instead of “their” be specific.

Page 38 – the term “dissociative amnesia” is bought in. What is meant by this. Are we talking about “inter-identity amnesia” or a DSM-5 diagnosis that is not DID? Same for mention of it on page 39.

Reviewer #2: This study assesses current gap of knowledge on autobiographical memory in DID patients. The study has a very strong design and by using two different methologies, authors try to extend the knowledge known about autobiographical memory in DID patients. They discuss their findings well and lead to a better understanding of the concept.

However, there are some parts of the study that are not clear.

1. In study 1, authors may define groups 3 and 4 in more detail. It is now well understood by the readers if these groups are also DID patients. What do they mean by saying non-clinical group and amnestic group?

2. Authors should define how they calculated the sample size for each group and the total number of participants. The number of participants in each group is lower than the number expected for representable power.

3. Tables should include the p value for each group statistical analysis, in addition to the descriptive numbers.

4. Very small number of stimulator participants were given DDIS which is a limitation.

5. In study 2, groups have a significant difference in age, in addition to small sample size for each group. This makes it difficult to generalize their findings.

6. PLOS authors have the option to publish the peer review history of their article (what does this mean?). If published, this will include your full peer review and any attached files.

Reviewer #1: No

Reviewer #2: No

---

## [Author Response · Author response to Decision Letter 0]

22 Dec 2020

Vedat Şar, MD 

Academic Editor, 

Plos One

20th December, 2020

Dear Vedat

Re: PONE-D-20-30695

Inter-Identity amnesia for neutral episodic self-referential and autobiographical memory in Dissociative Identity Disorder: An assessment of recall and recognition

PLOS ONE

Thank you for the opportunity to revise and resubmit our manuscript to Plos One. We have read the reviews carefully and appreciate the time and energy given by yourself and the Reviewers. Below we have outlined each comment in bold and then indicate how we have addressed the issues in our revision. As instructed we have included a clean and a tracked version of the revised manuscript. We feel the revisions have strengthened the paper and look forward to hearing from you. Rather than having the study protocols located at “Protocols.io” we have all our materials including data on an OSF site, which is accessed here: https://osf.io/u4r6k/. I send this letter and revised manuscript on behalf, and with the expressed permission, of my coauthors.

Yours sincerely

Martin J. Dorahy

School of Psychology

University of Canterbury

Private Bag 4800

Christchurch, New Zealand

Ph: +64 3 3643 416

Email: martin.dorahy@canterbury.ac.nz

Academic Editors comments

Thank you for this comment, we have updated the formatting to respect these guidelines.

2) We note that you have stated that you will provide repository information for your data at acceptance. Should your manuscript be accepted for publication, we will hold it until you provide the relevant accession numbers or DOIs necessary to access your data. If you wish to make changes to your Data Availability statement, please describe these changes in your cover letter and we will update your Data Availability statement to reflect the information you provide.

The data is publicly available at this OSF site, which we have made clear in the paper: https://osf.io/u4r6k/

Reviewer 1 comments:

3. The biggest problem I have is with the terminology “amnestic comparisons”, or simply “amnestic” (as compared to non-amnestic) in the abstract. The term needs to be clarified to highlight that these people are not amnestic, and that the process by which they have restricted access to information is not amnesia. Alternative wording – which I will leave up to the authors to suggest is needed along the lines of “amnestic simulators” or “partial information group”. Even identities in this context is confusing, and terms such as memory deficits given there was no memory.

Thank you for his comment, these groups have been re-named with the “amnesic comparisons” now named the “partial information comparisons” and the “non-amnesic comparisons” now named the “full information comparisons”. We agree that these terms better represent the conditions of the comparison group.

4. Page 5 – please clarify sentence as: “4) non-clinical comparison participants given stimuli shown to both identities (i.e., all of the stimuli; nonamnesic group).”

Thank you for this comment, the sentence has been clarified to indicate that the participants in this group were given the stimuli presented to both identities in the DID and simulator groups. It now reads:

4) non-clinical comparison participants given stimuli shown to both identities in the DID and simulator groups (i.e., all of the stimuli; full information group). 

5. Page 5 – reference to “distractor performance” needs more context, or left out until the discussion that describes this.

Thank you for this comment, the term has been removed and can be found following the description later in the paper.

6. Page 6 – footnote 1. is important and should be moved to main text, and further clarification. Were these trauma-knowing identifies assigned to A or B, or was there an attempt to evenly distribute them?

Thank you for this comment, we have now moved footnote 1 to the main text. We also added some further clarification so that the section now reads:

They self-selected the two identities (labelled A and B) that took part in the study with the instruction to chose one identity with more awareness of past distressing experiences and one identity with less awareness of autobiographical distressing experiences. They chose which identity took part first (as identity A) and second (as identity B), and awareness of distress was not explored in this study.

7. Page 8 – perhaps specify that Post-experiment Questionnarie determination of genuine vs feigned assessed all facets of the research, i.e. measures, experiments, overall conduct.

Thank you for this comment, this has been clarified on page 9. The section now reads:

 A post-interview questionnaire was included for the primary researcher to rate whether DID and simulator participants appeared to present with features of DID. The researcher completed a questionnaire for each participant to determine whether they appeared to have genuine or feigned DID symptoms (e.g., amnesia), which assessed multiple facets of their presentation such as whether they appeared to have distinct identities and in what ways their appearance differed across the tasks.

8. Page 10 - In regards to prompts, how is the number or prompts taken measured/accounted for?

The prompts in the FRImmediate were given to all participants, so they were not accounted for as there was no difference between them. For the FRDelayed, all participants got the first prompt, but not necessarily the other two. Unfortunately no recording was made of who did and did not get the latter two prompts. 

9. Page 11 – can you please provide more information on know / remember – is each dimension assessed or is it either/or? Is it a Likert scale? What if a respondent doesn’t have a clue if they know/remember the information of think it’s a bit of both?

We followed the standard procedure for the remember/know task, where participants either state if they remember or know the event occurred. If they remember they hit the R key on the keyboard, if they know they hit the K key, so it is a dichotomous rather than likert response. We hope we have made this clear in the section on the task that now reads: 

If participants indicated they recognised the sentence as representing a detail from the vignettes, they were asked to state whether they either ‘remembered’ or ‘knew’ that it had happened. A ‘remember’ response was explained to participants as being paired with an actual recollection of the event, while a ‘know’ response was explained as being paired with only a feeling that the event happened (Tulving, 1985). Participants pressed the R key on the keyboard to indicate ‘remember’, and the K key to indicate ‘know’. This task assessed autonoetic and noetic memory retrieval.

There is no provision in the task for ‘not having a clue’ or for saying both as participants are asked to decide which fits best for them, favouring the one that feels most accurate. We did not have a case where participants said they could not decide or it was both, but we acknowledge not enquiring about such options. 

10. Page 13 – I think the findings that the similarities between those with DID and simulators is where the information and coaching is, but the overall quality of the dissociative/amnestic experience being different needs to be highlighted in the discussion drawing on a range of findings in this paper, starting here. The paper really show that DID is a complex phenomenon that is incredibly difficult to feign. From their lower DES (I would expect simulators to score similarly high on the DES amnesia subscale, and depersonalization items that touch on being disconnected from the self, but far lower on absorption items). Differences in PTSD symptoms, differences in know/remember, memory being more porous in the DID samples, less likely to say for sure that ‘distractor’ events did not happen. I think there is important findings for the fantasy/trauma debate that need to be highlighted.

Thank you for this question. We have included a passage to highlight these differences more explicitly in the general discussion. 

Further, regarding comparisons with the simulator group, the DID sample not only showed differences in identity B recognition profiles, they also showed higher scores of dissociation and PTSD. They also indicated that they interpreted experimental stimuli as being significantly more negative emotionally. In terms of their presentation, DID participants were more likely to be rated as presenting with changes in appearance based on affect, body posture and voice and facial characteristics as well as the appearance of distinct dissociative identities. These results collectively suggest that DID is not easy to accurately feign, at least not by the current sample of professional and amateur actors. More specific samples (e.g., a selection of individuals high in fantasy proneness), might be worthy of investigation in future studies.

Of note, as we did not include reference to the DES subscales (amnesia/depersonalisation/absorption) we have not made additional reference as discussed in your comment above.

11. Page 21 - When we have p = 1.00 can we say the groups were identical, rather than not significantly different?

Thank you, we have now ensured this is no longer communicated by putting ‘significantly differ’ or words to that effect.

12. Page 21 – I am a little uncomfortable with the terms that imply “more liberal”, “more hesistant” “more conservative”. Is that really what is happening? I doubt we know. And if that’s the case then perhaps a “less likely”.

Thank you, we have followed your lead, and reworked the sentence as follows:

This suggests that simulators and amnesic comparisons were biased towards being less likely to indicate that they recognised sentences to represent events from the vignettes) at vignette 2. Comparatively, DID participants were more likely to bias towards recognizing sentences from vignette 2, but like the simulator and amnesic groups, were less likely to recognize these sentences than the nonamnesics.

13. Page 26 – I am uncomfortable with the term “failing to complete”. If one of the participants saw this, it would seem a harsh judgement. Perhaps “unable to complete”.

Thank you for this comment, this has now been changed to “unable to complete”.

14. Page 29 – Can we have exact data on “less likely to be rated as presenting with genuine dissociative identity and more likely to be feigning. Was this statistically significant?

Thank you for this comment, we have now included the relevant statistical data on page 29 where you can see that both analyses were statistically significant.

15. There is an issue with headings. We have discussions, then general discussion, and no conclusion.?

Thank you for this comment, these have now been updated to account for the appropriate formatting guidelines.

16. Page 37 – Line 7, sentence commencing “Their” – Instead of “their” be specific.

Thank you for this comment, this has now been changed to indicate that the DID group was the group being referred to.

17. Page 38 – the term “dissociative amnesia” is bought in. What is meant by this. Are we talking about “inter-identity amnesia” or a DSM-5 diagnosis that is not DID?

Same for mention of it on page 39.

Thank you for this comment, the term has been changed to “inter-identity amnesia” to allow for consistency throughout the paper.

Reviewer 2 comments: 

18. In study 1, authors may define groups 3 and 4 in more detail. It is now well understood by the readers if these groups are also DID patients. What do they mean by saying non-clinical group and amnestic group?

Thank you for this comment, please see point 3 where we have renamed these groups (now termed partial and full information groups) which we hope will also aid further clarification for this comment. 

19 . Authors should define how they calculated the sample size for each group and the total number of participants. The number of participants in each group is lower than the number expected for representable power.

Thank you for this comment. Given the challenges in collecting participants for the DID group that had the capacity to effectively switch between dissociative identities (which was a criterion for inclusion), the researchers did not engage an initial power analysis. Rather they followed previous experimental research (Huntjens et al., 2012; Marsh et al., 2018) and set out to include 20 participants for analysis.

In the discussion (p. 41), the following statement is included:

It is important to recognise the relatively small sample sizes, especially for the DID and simulator groups. Future studies should aim to include larger DID samples to replicate the current findings in a more powerful design.

20. Tables should include the p value for each group statistical analysis, in addition to the descriptive numbers.

Thanks for this suggestion. We have added significant p values to tables 2, 3 and 5 using alphabetical letters to show differences across groups to increase efficient presentation. Because the demographic tables contained information that could not be assessed with inferential statistics, we did not include a p values for these tables (1 and 4). 

21. Very small number of stimulator participants were given DDIS which is a limitation.

Thank you for this comment. This point has now been included within the text (see point 22 below)

22. In study 2, groups have a significant difference in age, in addition to small sample size for each group. This makes it difficult to generalize their findings.

Thank you for this comment. We have now included it within limitations section with this inclusion: 

In addition, the current studies present results from a relatively stable DID sample due to the requirements of participants being able to switch on demand and only a small number of simulator participants were administered the DDIS. In addition, because in study two the simulator group was significantly younger than the DID group, it is difficult to generalise these findings.

---

## [Decision Letter · Decision Letter 1]

11 Jan 2021

Inter-Identity amnesia for neutral episodic self-referential and autobiographical memory in Dissociative Identity Disorder: An assessment of recall and recognition

PONE-D-20-30695R1

Dear Dr. Dorahy,

We’re pleased to inform you that your manuscript has been judged scientifically suitable for publication and will be formally accepted for publication once it meets all outstanding technical requirements.

Kind regards,

Vedat Sar, M.D.

Academic Editor

PLOS ONE

**Comments to the Author**

1. If the authors have adequately addressed your comments raised in a previous round of review and you feel that this manuscript is now acceptable for publication, you may indicate that here to bypass the “Comments to the Author” section, enter your conflict of interest statement in the “Confidential to Editor” section, and submit your "Accept" recommendation.

Reviewer #1: All comments have been addressed

2. Is the manuscript technically sound, and do the data support the conclusions?

Reviewer #1: Yes

3. Has the statistical analysis been performed appropriately and rigorously? 

Reviewer #1: Yes

4. Have the authors made all data underlying the findings in their manuscript fully available?

Reviewer #1: Yes

5. Is the manuscript presented in an intelligible fashion and written in standard English?

Reviewer #1: Yes

6. Review Comments to the Author

Reviewer #1: (No Response)

7. PLOS authors have the option to publish the peer review history of their article (what does this mean?). If published, this will include your full peer review and any attached files.

Reviewer #1: No

---

## [Editor Report · Acceptance letter]

4 Feb 2021

PONE-D-20-30695R1 

Inter-identity amnesia for neutral episodic self-referential and autobiographical memory in Dissociative Identity Disorder: An assessment of recall and recognition 

Dear Dr. Dorahy:

I'm pleased to inform you that your manuscript has been deemed suitable for publication in PLOS ONE. Congratulations! Your manuscript is now with our production department. 

Kind regards, 

on behalf of

Dr. Vedat Sar 

Academic Editor

PLOS ONE